# MealWatcher: A New Toolset to Record Meals for the Development of Automated Energy Intake Estimation

Faria Armin[1], James Jolly[1], James Nguyen[1], Lakshmi Rangaraju[1], Daniel Regan[2], Zoe Brown[2], Sharon O'Toole[2], Hollie Raynor[3], Leslie Brick[4], Elissa Jelalian[5], J. Graham Thomas[5], Stephanie P. Goldstein[5], and Adam Hoover[1], *Senior Member, IEEE*

*Abstract*—**Classic tools for measuring energy intake, such as food diaries and 24 hour recalls, are burdensome to use and have significant measurement error. This hinders research and interventions in obesity treatment and comorbidities such as diabetes and heart disease. New tools are being developed to automate the measurement of energy intake, such as wearable devices like smartwatches. Towards this goal, several datasets have been collected and made publicly available that include hand motion. However, these datasets have been limited to the wrist part of hand motion, and have only been collected in controlled environments such as labs or cafeterias. In this work we describe a new toolset that supports data collection from a smartwatch and smart ring simultaneously, to be used in home, cafeteria, and free-living environments, with real-time user feedback to assist with energy intake estimation. This data collection is ongoing and will eventually encompass 600 subjects. This paper describes the toolset and preliminary results for 16 subjects, including a comparison of smart ring versus smartwatch intake gesture detection. The smart ring achieved an F1 score of 0.74 compared to an F1 score of 0.8 from the smartwatch. Finally, we describe the full set of experiments we intend to perform with the complete dataset.**

*Index Terms*—**smart ring, motion sensor, wearable device, gesture recognition, energy intake measurement**

## I. INTRODUCTION

Traditional tools to measure energy intake rely on self-reporting tools such as logging frequency of food consumption, food diaries, and 24-hour recalls of the foods consumed during the day [1], [2]. However, self-reported methods have a number of limitations, including high user and experimenter burden, interference with natural eating habits, and decreased compliance over time [3], [4]. There is a strong consensus in the research community that better tools are needed for energy intake measurement [5]. To address these shortcomings, wearable devices are being developed to automatically detect and measure consumption [6]. Examples including wrist-worn devices that can measure intake gestures (bites) [7], eyeglass and earpiece devices that can measure motions and sounds associated with mastication (chews) [8], and throat-located devices that can measure forces and sounds associated with ingestion (swallows) [9].

This paper describes MealWatcher, a new toolset to collect data for the development of smartwatch and smart ring devices for measuring energy intake. We focus on wrist and hand-located devices because of their preference in the general population; e.g. a survey of 96 subjects found that a smartwatch was preferred over smart glasses or a necklace for monitoring intake [6]. However, a smartwatch must be worn on the wrist of the dominant hand because it is the hand most commonly used for eating [10]. This goes against the cultural norm of wearing a watch on the wrist of the non-dominant hand [11]. Rings do not have a strong cultural bias to be worn on the non-dominant hand [12]. A main purpose of MealWatcher is to enable simultaneous data collection from a smartwatch and smart ring to compare accuracy for detecting intake gestures.

Table I lists all publicly available datasets containing hand motion for the research of intake gesture detection [10], [13]–[15]. Our new dataset will be the first to contain both wrist and finger motion as recorded by a smartwatch and smart ring worn at the same time during each meal. It will also be the largest of its kind, and will be made freely and publicly available upon the completion of its collection. We plan to collect data from 600 total subjects, with half of them eating a meal in a cafeteria setting and the other half eating a meal in their own home. We are doing this because many previous works have shown a decrease in accuracy in the recognition of intake gestures as experiments progress from laboratory to free-living environments [16]–[19]. One hypothesis is that intake gestures in-the-wild exhibit more variability than intake gestures captured under controlled conditions in laboratory environments [20]. We therefore plan to train two sets of classifiers, one on data collected in a cafeteria setting, and one on data collected in private home settings. This will allow us

This work is funded by NIH grant R01DK135679.

[1]Department of Electrical and Computer Engineering, Clemson University, Clemson SC 29631, USA (emails: farmin@clemson.edu, jpjolly@clemson.edu, jhn@clemson.edu, lrangar@clemson.edu, ahoover@clemson.edu).

[2]Weight Control and Diabetes Research Center, The Miriam Hospital, Providence RI 02916, USA (emails: DRegan@lifespan.org, ZBrown@lifespan.org, sotoole@lifespan.org).

[3]Department of Nutrition, University of Tennessee, Knoxville TN 37996, USA (email: hraynor@utk.edu).

[4]Department of Psychiatry and Human Behavior, Alpert Medical School, Brown University, Providence RI 02903, USA (email: leslie_brick@brown.edu).

[5]Department of Psychiatry and Human Behavior, Weight Control and Diabetes Research Center, The Miriam Hospital/Brown Alpert Medical School, Providence RI 02916, USA (emails: Elissa_Jelalian@brown.edu, John_G_Thomas@brown.edu, stephanie_goldstein@brown.edu).

TABLE I
COMPARISON OF OUR NEW DATASET USING MEALWATCHER
(COLLECTION IN PROGRESS) VERSUS EXISTING PUBLIC DATASETS OF
HAND MOTION FOR INTAKE GESTURE RECOGNITION.

| Dataset | Hand location(s) | Setting(s) | Total subjects | Total meals |
|---|---|---|---|---|
| ACE [13] | Both wrists | Lab | 7 | 13 |
| Clemson [10] | 1 wrist | Cafeteria | 276 | 276 |
| OREBA [14] | Both wrists | Lab | 100 | 100 |
| FIC [15] | 1 wrist | Lab | 12 | 21 |
| **This Work** | 1 wrist, 1 finger | Home, cafeteria | 600 | 600 |

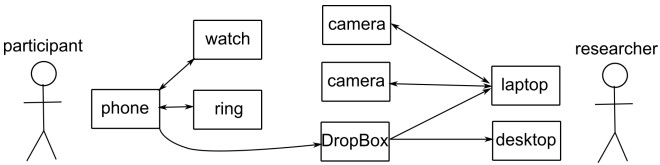

Fig. 1. Overview of all hardware and software pieces used during data collection.

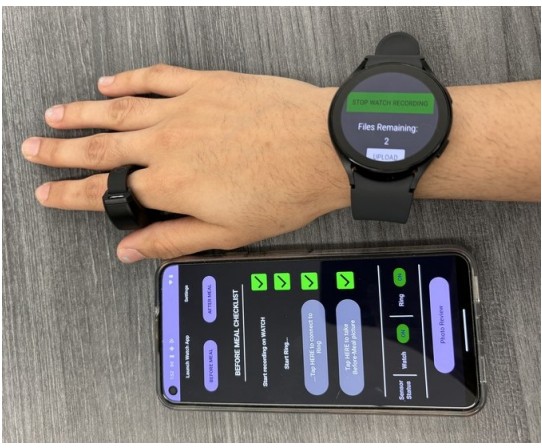

Fig. 2. Smart ring, smartwatch, and smartphone used for recording data with MealWatcher toolset.

to evaluate the effect of setting on intake gesture recognition and help guide future research into the collection of training data.

The novelty of this work can be summarized as follows:

- We describe a new dataset being collected that includes finger motion from a smart ring and wrist motion from a smartwatch, simultaneously recorded during meals.
- We describe methods to collect data in home and cafeteria settings to support the development and comparison of classifiers trained on data collected in different environments.
- We present MealWatcher, a toolset built to support this data collection.

Finally, we report a preliminary experiment on data collected from the first 16 subjects showing a comparison of intake gesture recognition from finger motion compared to wrist motion.

## II. MEALWATCHER: A NEW TOOLSET

The MealWatcher toolset is designed and developed to support several functions:

1) Collect motion data from a smart ring and smartwatch simultaneously during meals.
2) Allow deployment in 3 different environments: cafeteria, home, and free-living.
3) Collect brief survey data at the end of each meal to determine if simple questions can provide enough nutrition context about the beverages and foods consumed to improve energy intake estimates.

To support these goals, we built a toolset consisting of several components. Figure 1 shows an overview of our toolset. We will first discuss the workflow of the participants and then explain how the tools are used in that workflow.

### A. Participant Workflow

The participant wears a smartwatch and smart ring which are controlled through their smartphone from our MealWatcher app while eating meals (see Figure 2). They can use their phone for other activities while our app runs in the background. The phone uploads all data to Dropbox after each meal for remote storage and retrieval by the research group. During a single meal eaten in either a home (50% of participants) or cafeteria (50%), the researchers train participants on how

to use MealWatcher. The meals are observed by 1-2 cameras (GoPro) to video record these eating sessions from a comfortable distance (2 or more meters). During these observed meals, participants clap after cameras are started and before beginning to eat. These claps are later used to help synchronize the sensor recording files with the video recordings. The video recordings are then used for labeling the ground truth times of bite and drink gestures.

Following this training session, each participant is instructed to record meals for 4 days during everyday life (free-living environment). There is no video recording of these meals. Instead, this data will be used to evaluate energy intake estimates from the devices. For ground truth of energy intake, participants complete the ASA24 24-hour recall at the end of each day, which generates an interview-driven list of meals and food and beverages consumed in each meal [21]. The participant can take before and after meal photos in MealWatcher and then review these photos later to help complete the ASA24. Finally, the participant is instructed to complete a survey after every meal. The questions in the survey are designed to obtain nutritional context that may help convert intake gesture counts into estimates of energy intake (e.g. did the meal include a caloric or non-caloric beverage).

### B. Tools Overview

To support a diverse study population, we developed MealWatcher to run in both the Android and Apple ecosystems. The smartphone acts as a hub that communicates with both the smartwatch and smart ring. Both the Android and Apple smartphones have companion operating systems that run on

TABLE II
PROGRAMMING LANGUAGES AND IDEs USED FOR DEVELOPMENT OF
SOFTWARE COMPONENTS.

| Device | O/S | Prog. Lang. | IDE |
|---|---|---|---|
| Phone | Android | Java | Android Studio |
| Phone | iOS | Swift | XCode |
| Watch | Wear OS | Java | Android Studio |
| Watch | watchOS | Swift | XCode |
| Ring | - | Java, Swift | Android Studio, XCode |
| Cloud storage (Dropbox) | - | Java, Swift, C | Android Studio, XCode, MSVC |
| Desktop/Laptop | Windows | C | MSVC |

smartwatches, and their application programming interfaces (APIs) support integrated smartphone and smartwatch app development. We used these to develop companion MealWatcher watchOS and wearOS apps for recording the watch sensor data. The smart ring we chose for this project is the Genki Wave ring. It runs proprietary code on its microcontroller to operate as a Bluetooth peripheral. The MealWatcher phone app interacts with the ring by operating as a Bluetooth central device on the smartphone. Genki provides their Wave API to interface with their ring hardware. However, it only supports Python programming and was only intended to connect their ring hardware to desktop and laptop computers. We therefore rewrote this library, porting it to both Java (Android) and Swift (Apple). Additionally, we developed custom dashboard software to record video and provide synchronized playback and labeling of all data. The operating systems, programming languages, and integrated development environments (IDEs) used to develop all the software components are summarized in Table II. It took our group 6 months to develop this software and included 5 members working full-time on the programming.

### C. MealWatcher Phone and Watch Apps

The MealWatcher phone app design went through several iterations during development and testing. A goal common to all our designs was to simplify the number of operations that a participant has to complete to operate everything. In an ideal situation, a single button press could start and stop recording on all devices. However, the smart ring requires manually pressing a button to start a data streaming session, and the Apple ecosystem does not allow the smartphone to fully control smartwatch apps (the user must manually wake the watch before a watch app can be started). During testing of various designs, we found that users could be confused by the various actions they needed to complete, especially if devices struggled to communicate state conditions with each other. In some cases, users would repeatedly try turning sensors on and off from one device (the phone) instead of realizing the other device was in a wrong state (the watch was asleep).

Therefore, we came up with a design we call "4 taps". The participant is instructed to perform 4 taps before starting and after finishing each meal. Figure 3 shows the design, which

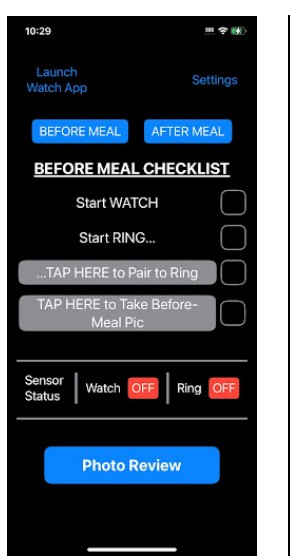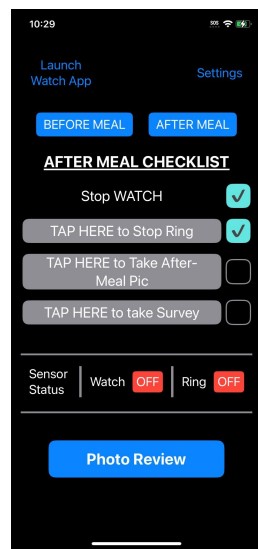

Fig. 3. MealWatcher's 4-taps design, in which the user is asked to complete 4 actions before and after eating a meal. This interface requires the user to interact with each of the devices (watch, ring, phone, camera).

emphasizes that the user should complete a series of 4 actions before starting to eat a meal: (1) start watch sensors, (2) power the ring, (3) start recording on the phone, and (4) take a picture. As each action is completed, a check mark is placed next to the instruction on the app. Moreover, there is further feedback to ensure whether the watch and ring sensors have started to record data or not with the green/red color of the sensor status. Upon finishing a meal, the user is again asked to execute 4 taps: (1) turn off the watch sensor by tapping a button in the watch app, (2) stop the ring recording (the ring automatically powers off in approximately 30 seconds), (3) take an after-meal picture, and (4) complete the meal survey.

The user can take multiple before- and/or after-meal photos if they choose to do so. This can be useful if a user consumes multiple servings or wants to document multiple dishes in different photos. These photos can be used by the participants during completion of the ASA24. Also, the researcher can use these photos to help assess if the participant is following the workflow correctly.

There is also a settings section in MealWatcher, where a unique participant ID (PID) will be set by the researcher, which will generate filenames for all data recorded by the participant and ensure user anonymity. Moreover, this section allows the phone to be paired with a unique smart ring so that multiple participants can operate in the same room. Finally, we include some convenient actions in the settings, such as to force-upload all current data to Dropbox. This function can be helpful when cellular data or internet access are not available during a meal and that data must be uploaded later.

The MealWatcher watch app has a home screen with a single button that displays the current sensor status and can be used to turn the recording on and off. This watch app is needed to control and read the motion data on the watch using the local sensors. Upon completing a recording, MealWatcher

transfers the saved data file to the phone. The phone and watch operating systems handle device pairing and provide an interface for the two devices.

### D. Data dashboard

For reviewing and labeling the recorded meals, we have developed a custom dashboard. It supports several functions, including video playback with a plot of synchronized sensor data. Figure 4 shows some examples. The vertical green bar in each plot of sensor data shows the time currently displayed in the video. Synchronization of the video and sensor data is done by searching for a clap in the video and separately finding the clap motion in the sensor data. Figure 4(a) shows an example after synchronization has been completed. The participant is asked to remain motionless for 3 seconds before and after the clap to make it easier to find the clap motion. We use frame-by-frame review of the video and sensor data near the transition from rest to motion (starting the clap), and from motion to rest (ending the clap), to obtain the highest possible accuracy in synchronization.

The dashboard software also supports labeling of intake gestures. Figure 4(b) shows an example. Note that the camera was positioned about 2 m from the participants to facilitate a comfortable experience. In figure 4(b), the video has been digitally zoomed to a smaller window to provide a clearer perspective on participant actions. The figure shows a moment when intake is about to occur. The label associated with this intake is graphed as a purple vertical line over the sensor data. All intake gestures are manually labeled with the hand(s) used (left, right, both), utensil used (hand, fork, spoon, knife, chopsticks), dishware (plate, bowl, glass, cup), and food/beverage items consumed during the intake gesture. This list of food items is unique to each meal and is created based upon information collected during a brief interview of participants about their meals.

### E. Data collection suitcase

Data collection is taking place in numerous locations. In the cafeteria setting, we are setting up all equipment independently each time we meet a group of 4 participants to train them. We chose the cafeteria for a data collection site instead of a laboratory setting to provide access to hundreds of different types of food items and beverages. We loan smartwatches and smart rings to the participants for the 4-day duration of their free-living data collection, and break down the rest of our equipment for return to our lab. In the home setting, we plan to visit 80 unique homes, where the same process of equipment setup, training, loan, and break down will be completed. We therefore packaged all our equipment into several "data collection suitcases". Figure 5 shows an example. Each suitcase contains 8 watches (4 Android, 4 Apple), 4 rings, 2 cameras, and a laptop for data recording. It also contains watchbands of various sizes (if needed), charging cables, and other support items. These suitcases help us transport this large number of small items needed for data collection.

## III. PILOT EXPERIMENT: COMPARING FINGER VERSUS WRIST MOTION FOR RECOGNIZING INTAKE GESTURES

In this section we show an experiment using our preliminary data to compare wrist versus finger motion for detecting intake gestures. This helps demonstrate how our new dataset expands possible experiments beyond what current publicly available datasets can support. We show a benchmark test of previously published intake detectors on our pilot data. We also run these same detectors on two large, publicly available intake detection datasets with wrist motion data. These benchmark datasets provide a reference for how our pilot data results compare to public datasets used in current state-of-the-art models. While the wrist motion data from our pilot data will be directly comparable to these public database, the finger motion data is a novelty to the pilot data, and comparisons to other datasets provide insight into this new sensor modality.

### A. Benchmark Datasets

We use two public datasets for benchmarking. We chose the two largest because they best support training neural network classifiers which can require a lot of training data. The first dataset is the Clemson dataset, collected on 276 participants in a public cafeteria setting. This dataset has a lot of variety, with 374 different food and drink items eaten across the meals with many different utensils. This dataset was recorded at 15 Hz using a custom, wrist mounted IMU device that tracked the dominant hand of participants.

The second dataset is the OREBA Intake dataset, which was collected in a lab with 2 to 4 people eating around a table together. This dataset contains subsets; the first subset is the OREBA-DIS which contained discrete portions served before eating, and the second is the OREBA-SHA which had a shared dish at the table from which all participants served themselves. In this paper we use the OREBA-DIS due to its similarity to the cafeteria setting matched by our pilot dataset and the Clemson dataset. OREBA-DIS contains 100 meals from 100 different participants, who all ate the same meal provided in the study. OREBA recorded both wrists using IMU devices and recorded at 64 Hz. To provide equitable comparison to other datasets, we are only using the dominant hand data in OREBA in this study.

### B. Detection Methods

We tested two different intake gesture detectors, one that uses a heuristic algorithm and one that uses a neural network. The heuristic-based classifier relies upon a single feature that measures the roll of the wrist [22], which has been shown to be locked to the roll of fingers on the same hand [12]. Therefore we anticipate it should do well on our finger data without requiring retraining of the classifier. The neural network classifier we chose is representative of state-of-the-art for detecting intake gestures from wrist motion [23].

The heuristic algorithm uses a combination of rotation and timing thresholds to detect the lifting of the wrist and rotation of the hand toward the mouth and back to rest. The detector first identifies a positive rotation of the wrist (e.g. rotating

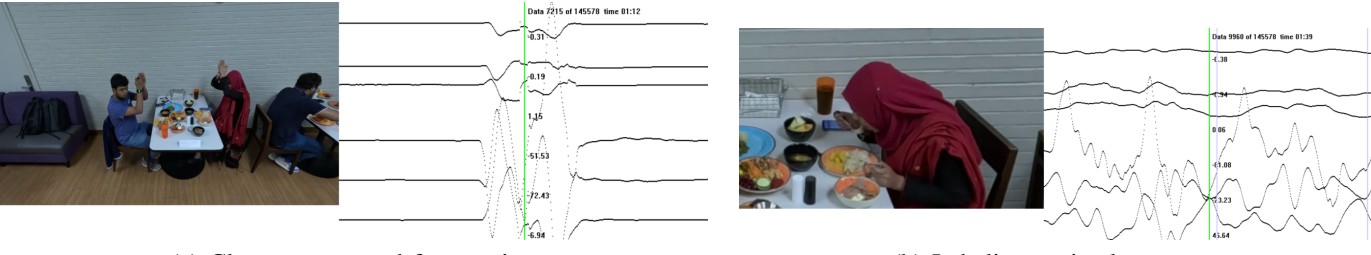

(a) Clap gesture used for syncing        (b) Labeling an intake gesture

Fig. 4. Our dashboard software for reviewing recorded data. The 6 signals plotted on the right side are accelerometer X, Y, Z and gyroscope X, Y, Z over time. The vertical green bar in the sensor data is synchronized to the video display.

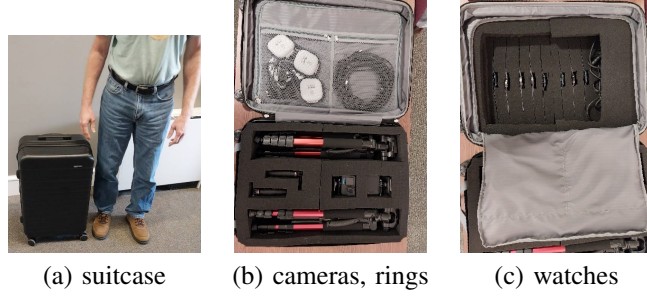

(a) suitcase    (b) cameras, rings    (c) watches

Fig. 5. Data collection suitcase used to deploy equipment in cafeteria and home settings.

palms from facing downward to facing up) to identify a person picking up food and bringing it to their mouth. The model then look for a negative roll to identify the returning of the wrist from the intake back to a neutral position. These roll events must be spaced out by two time thresholds to prevent excessive triggering of detections. Values used in the original paper looked for rotations of greater than 10 deg/sec for each direction, with start and ending of detections being spaced out by 2 seconds and detections having a minimum spacing of 8 seconds. See [22] for pseudo-code and other details of how this detector was created.

The second detector is a re-implementation of the CNN-LSTM network in [23]. This model slides a 2 second window across the data outputting the probability of the window being an intake gesture, then post-processing these probabilities to create detections. To create this CNN-LSTM detector, we followed training parameters specified in the original paper. All data was resampled with cubic interpolation to match the original 64 Hz data rate for which this model architecture was designed. The data passes through 4 convolution layers, each with 128 filters and each successive layer increasing filter sizes, with respective filter sizes for the layers being 1, 3, 5, and 7. The model then flows into two LSTM layers, each with 64 memory units. All CNN layers use a ReLU activation and all LSTM layers use hyperbolic tangent activation. Finally, the model ends with a dense layer to produce a output targeting the class for each input. An Adam optimizer and default learning rate are used. See [23] for additional details.

Because this detector requires training, we trained a separate version of the model for each of the two public datasets.

Both datasets are large and have sufficient data sizes to train. We used an 80/20 training and testing split on both datasets. When evaluating this detector on our pilot wrist data, we used the OREBA-trained model and transformed our input data to match the format of the OREBA dataset. This choice was made since our pilot data is still small and would be prone to overfitting. Since there is no large public ring dataset, we used the same OREBA wrist trained model on our pilot finger motion data.

### C. Evaluation Metrics

To evaluate the detectors performance on our pilot dataset and the benchmark datasets, we match the detections to the ground truth intake gestures use the matching scheme initially described in [22]. This method scans the area around each detection, bounded from the preceding detection to the succeeding detection, and matches the detection to the closest unmatched ground truth. If each match counts as a true positive (TP), and if there are no unmatched ground truths within the scan window then the detection is marked as a false positive (FP). After attempting to match each detection, any unmatched ground truth intakes are marked as false negatives (FN).

With these matches, we calculate recall, precision, and F1 score. Recall, or true positive rate, is the percentage of the ground truths which are true positives, and is calculated using Equation 1. Precision, or the positive predictive value, is the percentage of detections which are true positives and is calculated using Equation 2. F1 score is the harmonic mean of these two values, and is the best single metric which balances recall or precision, preventing a model from under predicting to raise precision or over predicting to raise recall. Calculating F1 score is shown in Equation 3.

$$TPR = \frac{TP}{TP + FN} \tag{1}$$

$$PPV = \frac{TP}{TP + FP} \tag{2}$$

$$F1 = 2 \times \frac{PPV \times TPR}{PPV + TPR} \tag{3}$$

To achieve balanced results between true positive detections and the absence of false positive detections, we have tuned one threshold value in each of the detectors. For the Heuristic

Detector, we have adjusted the minimum time between predictions, with best results ranging from about 6-8 seconds. For the CNN-LSTM Detector, we have tuned the minimum threshold needed to trigger a detection.

## IV. RESULTS

This section first describes the data we have collected to date. We next describe results comparing two intake gesture detectors on our new data compared to benchmark datasets. Lastly, we describe preliminary comparisons of finger versus wrist motion for recognizing intake gestures.

### A. Dataset

Data is being collected under IRB2023-0146 at Clemson University with sIRB oversight to The Miriam Hospital at Brown University. The observed meal at the first site is being recorded in a cafeteria setting, while the observed meal at the second site is being recorded in home settings. All subjects provided informed consent to collect data. Subjects also provided informed consent to share videos collected in a cafeteria setting, but videos collected in home settings will remain confidential and only be used for labeling the times of intake gestures in sensor motion data.

At the time of this writing, steady-state data collection is in progress at the first site, and still undergoing internal testing and refinement at the second site. A total of 16 subjects have completed data collection. Each subject recorded one observed meal in a cafeteria setting, and 182 total meals during free-living (approximately 11 per subject).

In order to monitor the reliability of the data, we have developed a custom software dashboard. The dashboard automatically groups all files of a meal (5 total) which helps us to check whether a user completed all steps. We check if the begin/end timestamps of the watch and ring sensor recordings match the timestamps of the pre- and post-image files. In the survey of the MealWatcher app, we ask a few questions related to the quality of the data, specifically (a) whether the watch and ring work correctly or not, and (b) whether the participant was using their phone for other activities while recording the meal (anticipating that this could interfere with data quality, for example if the user accidentally closed the MealWatcher app). By observing these pieces of information from the dashboard, we can monitor the data quality.

Of the 16 observed meals, 2 had technical errors, and of the 182 free-living meals, 15 had technical errors and 14 had user errors. Technical errors were due to lost Bluetooth connections and app dozes (automated battery saving features built into smartphones). User errors were due to subjects forgetting to start a device before eating, or closing the MealWatcher app during eating (e.g. one user self-reports accidentally closing our app while switching to a web browser to watch a video during eating). Based on a review of this first 2.5% of the planned data to be collected, we have lightly refined MealWatcher to assist subjects with collecting good data, such as notifying them when Bluetooth connections are lost.

TABLE III
CHARACTERISTICS OF 1,141 TOTAL INTAKE GESTURES IN 14 OBSERVED CAFETERIA MEALS.

| Variable | Observed values |
| --- | --- |
| Hand used | 126 left hand, 995 right hand, 20 both hands |
| Utensil used | 415 hand(s), 298 fork, 1 knife, 404 spoon, 23 chopsticks |
| Food ratio | 1,036 food, 105 beverage |
| Food types | 84, including banana, burger, carrots, chicken, french fries, grapes, muffin, pasta, rice, and salad |

TABLE IV
MODEL PERFORMANCE OF HEURISTIC DETECTOR [22] AND CNN-LSTM DETECTOR [23] ON LARGE PUBLIC DATASETS AND ON OUR NEW PILOT DATA.

| Dataset | Heuristic detector | | | CNN-LSTM detector | | |
| --- | --- | --- | --- | --- | --- | --- |
| | F1 | PPV | | F1 | PPV | TPR |
| OREBA | 0.811 | 0.829 | 0.794 | 0.804 | 0.767 | 0.848 |
| Clemson | 0.820 | 0.811 | 0.830 | 0.784 | 0.751 | 0.821 |
| new (finger) | 0.800 | 0.791 | 0.810 | 0.740 | 0.749 | 0.730 |
| new (wrist) | 0.811 | 0.808 | 0.814 | 0.801 | 0.807 | 0.794 |

In the 14 observed meals without technical error, we annotated 1,141 total intake gestures. Characteristics are summarized in table III. The observed intake gestures exhibit great variety including the hand used, food/beverage ratio, utensil used, and food types. This can be contrasted against other datasets. In OREBA, only two different fixed meal choices were available to subjects, and only forks, knives and spoons (no hands or chopsticks) were used as utensils, in order to simplify data collection procedures [14]. The FIC dataset had the same utensil limitations, although subjects were free to choose their own foods [15]. The Clemson dataset is the most similar to ours, with all eating characteristics unconstrained and a broad range of food and beverage choices [10]. The variety of characteristics in this early data gives us confidence that food selections and eating styles are not being prohibitively restricted by MealWatcher and our methods used for data collection.

### B. Intake Detection Performance

Result metrics for intake gesture detections can be found in Table IV. The heuristic detector achieved similar performance (appx 0.80 F1) on all 4 datasets. We believe this is because the heuristic detector relies upon a single dominant feature, roll of the wrist/hand/forearm, that can be measured equally at the wrist and finger [12].

The CNN-LSTM detector achieved similar performance (0.80 F1) on our new wrist motion data, when compared to the two large public datasets (0.78 to 0.80 F1). The detector achieved lower performance (0.74 F1) on our new finger motion data. We believe this performance loss is because the model was not trained on any finger data, but rather trying to interpret the finger motion as wrist data.

The similarity of the watch pilot dataset to both public datasets supports that the finished dataset could be used as a

| Dataset | Heuristic Detection | CNN-LSTM Detection |
|---|---|---|
| Both Sensors Agree | 82.6 | 69.5 |
| Watch Detects Only | 9.6 | 19.5 |
| Ring Detects Only | 7.8 | 11.0 |

new benchmark for intake detection. The similar performance indicates that the varied foods in the cafeteria would be of a similar difficulty to foods in large, public datasets.

The similar performance of the ring pilot study to the wrist motion in the heuristic detections shows that some motion used to track intake (such as the rotation of the wrist) can still be tracked through the motion of the finger. This provides promise for potential uses of this new sensor location. And while the performance of a neural network model in this preliminary study demonstrates that models trained on wrist motion may not find the same features in finger motion data, we believe that these differences can be learned by a neural network once we have sufficient finger motion data on which to train a model.

*C. Finger versus Wrist*

To evaluate the differences in wrist motion detections compared to finger motion detections, we analyzed which intake gestures were detected at both hand locations. This highlights if certain intake gestures are better detected at the wrist compared to the finger, or vice versa. Results of how the two sensor locations agree or disagree in matching detections are summarized in Table V. For the heuristic detector, 82.6% of intake gestures were either found or missed by both detectors, demonstrating a high rate of agreement. Additionally, the remaining 17.2% of intake gestures that were found at one sensor location only were evenly split between the finger and the wrist, indicating that neither the finger nor the wrist motion had a strong advantage over the other in detecting intake. This high agreement demonstrate that some characteristics used to identify intake, such as the roll used in the heuristic detection, are clearly identifiable at both locations.

The CNN-LSTM detector had lower agreement, with only 69.5% of intake gestures being detected or missed by both the finger and the wrist data. When looking at the remaining 30.0% of gestures that were found at one sensor location only, the wrist had almost double the amount of detections (19.5% versus 11.0%), showing a stronger performance at the wrist compared to the finger. This disparity between finger and wrist detections for the neural network detection indicates that there are some features found in the wrist motion the model has learned which are absent in the finger motion. This distinction does not necessarily mean that wrist motion is better for detecting intake. Rather it highlights that there are some characteristics which distinguish wrist and finger motion during intake, and these differences can be used by models. With a large finger motion dataset to learn from, it is possible that a model could perform better using characteristics distinct to finger motion.

## V. DISCUSSION

In this paper we described MealWatcher, a new toolset designed to support collecting data to build automated tools for measuring energy intake. It enables simultaneous recording of finger and wrist motion and can be used in cafeteria, home, and free-living environments. It allows users to take before and after photos of their meals. It implements a survey to ask questions about the meal; the survey contents can be easily modified. MealWatcher is currently being used to collect data from 600 participants, with 16 complete at the time of this writing. Preliminary results indicate that MealWatcher supports collecting natural eating behaviors, and that a smart ring detects intake gestures at near equal accuracy to a smartwatch.

In the future, we plan to study several questions. First, after sufficient data is collected, we will train new neural network classifiers on finger motion to retest their accuracy. This will allow for a more fair comparison between smart ring and smartwatch devices. Second, we will train new classifiers on data collected in home settings versus classifiers trained on data collected in a cafeteria setting. This will allow us to evaluate the effect of environment, and specifically to determine if a cafeteria setting provides enough variability in eating behaviors to generalize to home settings. Third, we will evaluate the accuracy of energy intake estimation from both devices (per meal) by comparing their results to those obtained from ASA24 hour recalls. We do not have video recordings of the free-living meals and thus cannot evaluate the accuracy of intake gesture detectors. However, by combining the detected count of intake gestures with estimates of kilocalories per bite and kilocalories per drink, we can calcualte a total energy intake for the meal, and compare that against the ASA24 measurement for the same meal. This will give us arruacy estimates of the entire toolset.

Energy intake (EI) estimation is the ultimate goal (measure) of this area of research [24]. Previous work [25] has shown that EI can be calculated as the product of total intake gesture count and a scale factor called kilocalories per bite (KPB). This calculation assumes an average amount of kilocalories is consumed in each bolus. Demographic variables affect KPB, most notable gender, with females averaging 11 KPB and males averaging 17 KPB [26]. Age, height, and weight also have a modest impact on KPB [27]. The data collected by MealWatcher will continue to advance the science of these calculations. First, we will develop new classifiers capable of recognizing bite and drink gestures independently [28]. This will allow us to test formulas using the product of KPB and bite gestures plus the sum of KPD (kilocalories per drink) and drink gestures. Second, we will explore the impact of modifying KPB and KPD according to responses from the brief survey administered at the end of each meal concerning food and beverage content and secondary activities.

Dietary composition, secondary activities, and restricted food consumption can all have an impact on total energy intake [29]–[34]. Therefore, these survey questions are intended to determine dietary composition indicators, such as

fat, vegetables and sugar, whether participants were involved in other activities while eating, and to assess their eating pace. For example, if a user responds that they consumed a noncaloric beverage, then their KPD could be modified to zero. Therefore, we plan to explore the effect of the answers to these survey questions as modifiers upon KPD and KPB. The most impactful questions will be used to build a final classification model for future use.

When the planned data collection is complete, we will make of it publicly and freely available. It will be the largest dataset of its kind and will support the study of many research questions beyond those outlined here. We will also make all our code open source, publicly and freely available. We spent a great deal of effort designing and testing the MealWatcher toolset to support this data collection, and we thank the dozens of anonymous testers that have participated.

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
