# OpenReview forum: "MealWatcher: A New Toolset to Record Meals for the Development of Automated Energy Intake Estimation"
_IEEE.org/EMBS/BHI/2024/Conference — IEEE BHI'24_

### Official Review · Reviewer_thxs · 2024-08-10
**The paper provides a good contribution by proposing a new food intake gesture database with novel finger gestures.**

**Overall Rating:** 7
**Confidence:** 3

**Other Quality Metrics:**

(a) Clarity of writing: excellent
(b) Clinical Significance: good
(c) Methodological Novelty: good
(d) Experiments and Results: excellent

**Questions For The Authors:**

I don't have questions for the authors.

**Strengths:**

1. The interface between the subject and the smartphone app is easy to understand.
2. The introduction of finger motion data is novel in this area of research, and the database of finger motion would be unprecedented.
3. The acceptability of the proposed database is demonstrated by applying 2 different types of classifying methods, the heuristic detector and the CNN-LSTM detector, to both existing database and the proposed database.

**Summary Of The Paper:**

The paper proposes a new food intake gesture database with conventional wrist gestures and novel finger gestures. A series of software have been developed readily for participants to collect the data in various scenarios. A small portion of the data has been collected and verified with 2 types of methods, the heuristic detector and the CNN-LSTM detector, to demonstrate the acceptability of the proposed database. The novel finger data has the potential to provide better classification results if a model could be trained uniquely.

**Weaknesses:**

The limitation of existing databases would also be applicable to the proposed database, which is that the energy intake per gesture would only be accurate when the food is constrained to a predefined range.

---

### Official Review · Reviewer_Ugkp · 2024-08-12
**Energy intake tool that collects data from smartwatch and smart ring is propsed. The paper includes only preliminary results for 16 subjects.**

**Overall Rating:** 6
**Confidence:** 3

**Other Quality Metrics:**

(a) Clarity of writing: Good
(b) Clinical significance: Good
(c) Methodological novelty:Good
(d) Experiments and Results:Fair

**Questions For The Authors:**

I could not figure out how the reliability of data will be guaranteed. This seems to be a problem.

**Strengths:**

A tool to control the energy intake would be versatile for large number of population, and the proposed tool aims this. The paper is well written explaining the steps in detail.

**Summary Of The Paper:**

To follow up the energy intake, a tool is proposed. The tool supports data collection from a smartwatch and smart ring simultaneously, to be used in home, cafeteria, and free-living environments, with realtime user feedback to assist with energy intake estimation.
The presented results covers data from 16 subjects but the authors aim to collect data from 600 subjects. The process of data collection is explained in details.

**Weaknesses:**

The data collection suitcase seemed to be difficult to handle. There may be possibilty to use already set up tools.

---

### Official Review · Reviewer_vbMP · 2024-08-28
**MealWatcher toolset and study**

**Overall Rating:** 8
**Confidence:** 4

**Other Quality Metrics:**

(a) Clarity of writing: Excellent
(b) Clinical Significance: Excellent
(c) Methodological Novelty: Good
(d) Experiment and Results: Excellent

**Questions For The Authors:**

If the code for the toolset or the dataset is planned to be open sourced at the end of the study, that should be included in the paper.
In the classification task, is all free-living data being classified? It is unclear whether the goal of the project is to detect intake gestures during known meal events, or whether the goal is to detect eating gestures out of an entire day.

**Strengths:**

* The design choices for the toolset, as well as challenges faced during development and pilot study are great details that provide useful information for future research.
* The included figures and tables very descriptively show each important aspect of the project/results.
* Annotating labels with cameras and time synchronization are a great method for a strong ground truth.
* The reported performance metrics, including F1, PPV, and TPR, are an excellent set of metrics for the given task, considering the expected class imbalance.

**Summary Of The Paper:**

This paper presents the tools developed for, and preliminary results from MealWatcher study. The toolset includes an application that is compatible with both Android and iPhone devices, and which interacts with a smartwatch and smart ring. The dataset is recorded in both a controlled cafeteria setting as well as free-living conditions, and cameras are used to relay the ground truth. This paper also presents a heuristic-based approach and a neural-network based approach for detecting eating events.

**Weaknesses:**

* Figure 5a is not an image taken by the authors, it should be cited or replaced with an original image.
* Table IV's "Heuristic detector" shows "F1" "PPV" "PPV", but I think this third column is supposed to be "TPR".
* The paper suggests that there is class imbalance by presenting TPR, PPV, and F1 as the target metrics, but it would be nice to know exactly what the class imbalance is.

---

### Decision · Program_Chairs · 2024-09-23

Accept